# Distribution and Physiology of *Juniperus seravschanica* Trees in the Genow—The Southernmost and Arid Habitat of Iran

Abdolrahman Rahimian Boogar [1,*], Hassan Salehi [2] and Esmaeel Seyedabadi [3]

1 Department of Horticultural Science and Landscape Engineering, Faculty of Agriculture, University of Zabol, Zabol 98613-35856, Iran
2 Department of Horticultural Science, School of Agriculture, Shiraz University, Shiraz 71441-65186, Iran
3 Department of Agronomy, Faculty of Agriculture, University of Zabol, Zabol 98613-35856, Iran
* Correspondence: a.rahimian@uoz.ac.ir; Tel.: +98-543-1232-126

**Abstract:** *Juniperus seravschanica* is the southernmost population of *Juniperus* that has a limited habitat in the world near the equator. In Iran, the lone habitat of this species in the Genow mountains has been endangered with thin foliage, abscissing needles, and dried shoots. The current study investigated the effects of climatic, genetic factors, and physiologic indices on the distribution of *J. seravschanica*. Distribution was evaluated for 450 ha and physiological indices were evaluated for two groups: (A) trees with dried branches and (B) trees without dried branches. Results showed that the distribution of *J. seravschanica* in the Genow habitat was influenced by elevation, slope degree, aspect, and distance to stream. Results also indicated that max temperature and precipitation are two effective factors that have the highest effects on falling needles and drying branches of *J. seravschanica*. Chlorophyll, relative water content (RWC), and relative turgidity (RT) are significantly influenced by max temperature. Endangered trees with dried branches had a lower chlorophyll content, RWC, and RT than trees without dried branches. Vulnerability of *J. seravschanica* was significantly influenced by its genetic structure. Results of AMOVA showed 83% genetic variability between two groups of *J. seravschanica* trees.

**Keywords:** climate change; conservation programs; distribution maps; ecological landscape





## 1. Introduction

Genow is the most important mountainous region, with a unique ecosystem, near the coast of the Persian Gulf in southern Iran. There is high biodiversity of ornamental, medicinal, and animal species in this habitat, which creates a most attractive ecological landscape for tourism [1–3]. Climatically, the Hormozgan province has a dry and warm climate, but the climatic factors at the peak of the Genow mountains are lower than the surrounding lowlands [4].

*Juniperus seravschanica* is an evergreen species of conifers, distributed in Asia in Kazakhstan, Kyrgyzstan, Afghanistan, Pakistan, Iran, and Oman [5,6]. Forests of *J. seravschanica* have widespread distribution in the south and southeast of Iran, including the natural habitats of Kharmankohe in Fars, and Kohbanan, Khabr, and Rabor in Kerman [7,8]. Therefore, the southernmost populations of *Juniperus* sp. have an isolated and fragmented habitat far from the habitats of the species located in the Genow mountain [9]. *Juniperus* sp. has been used by local human as fence plants to protect crop field, and collected their wood for building and fuelwood [10]. Moreover, *Juniperus* sp. has various usage in traditional medicine according to their activities on anti-inflammatory, antioxidant, antimicrobial, anticholinesterase, and antirheumatic, and they are used to treat or cure, chronic arthritis, gout, and diabetes [11,12]. Also, some species of Juniper with more drooping foliage, and their resistance to drought and water deficit, are suitable for cultivation as an ornamental shrub and tree in urban and ecological landscapes [13–15]. In the present decade, drought extension and climate change have led to a recession of *J. seravschanica* habitats in semi-arid

or arid zones [16,17]. Moreover, in southern arid and semi-arid areas in Iran, the max and min temperatures are the most effective factors in the habitat suitability of junipers [18]. Other studies showed that the habitat suitability of *Kirengeshoma koreana* is influenced by soil profundity, water availability, and light intensity in the floor of the forest in the Mt. Baek-un region, South Korea [19]. Genetic diversity is an important factor in species adaptations against climate change and abiotic stresses [20,21].

Climate changes often along with drought, heat, and salinity stresses have intense effects on plants' physiological functions [22–24]. Relative water content (RWC) is generally influenced by stress and it is a better indicator of water stress for conifers [25]. Furthermore, *Juniperus przewalskii* has a different pattern of transpiration due to seasons and vapor pressure [26]. Moreover, the study of *Picea abies* validated that drought stress causes conifer xylem pressure loss [27]. Water stress causes xylem cavitation to occur in the leaves, roots, and stems of plants, and many studies investigated the susceptibility of plant tissue to cavitation and its impact on plant durability and productivity [28,29]. Chlorophyll content is another indicator of stress impacts on plants that is generally used for the investigation of plant behavior under stress conditions [30].

However, in recent decades, the increase in wrongful human activities, quick industrialization, and climate change have led to a threat to plants in natural habitats [31,32]. Hence, conservation of rare species and their habitat is very important for protecting the biodiversity and range of landscapes to improve the life, work, and play environments for human societies [33]. Furthermore, protecting genetic diversity is an essential approach to conserving forest trees in ecological landscapes. In this regard, the gene flow within and among populations has a significant effect on the outlines of genetic variation [21]. Moreover, pollen movement is recognized as a very effective factor for conserving genetic variation within and between populations and their relationship in natural forests [34,35].

Genow is the southernmost habitat of *J. seravschanica* in Iran, which is influenced by climate change. A previous investigation into the changes in hydro-climate factors of the past 30 years was performed on the southernmost basin of Iran (36), which found that the Genow habitat had an increase in average temperature (0.50–0.75 °C), and decreases in precipitation (75–82 mm/year), runoff (1–20 mm/year), and evapotranspiration (1–25 mm/year), and also a reduced amount of water storage (40–41 mm/year) during the period 1986–2016 [36]. In this study, this habitat was investigated through three major aims: first, to recognize the distribution pattern of endangered trees of *J. seravschanica* in the Genow habitat, second, to investigate the importance of climatic factors affecting physiological indices of *J. seravschanica*, and, third, to assess the genetic diversity within and among the population for endangered and healthy trees.

## 2. Materials and Methods

### 2.1. Study Area

The growing site of *J. seravschanica* is located in the Genow area in Hormozgan province, with geographical features at 56°5′58″ E and 27°23′42″ N. The Genow habitat is the lone ecological landscape near Bandar Abbas city, and its biodiversity is influenced by conditioning factors and local human activities. The current study covers approximately 450 ha of the natural habitat of *J. seravschanica* in Genow. The studied area has a moderate climate, with a long-term annual average temperature of 34–35 °C and rainfall of 50–100 mm (http://www.irimo.ir (accessed on 20 August 2021)). Topographically, the elevation of the investigated site area ranges from 1480 to 2291 m above sea level according to the digital elevation model (DEM) of the study area, while slope degrees range from 0 to 68.82° (Figure 1).

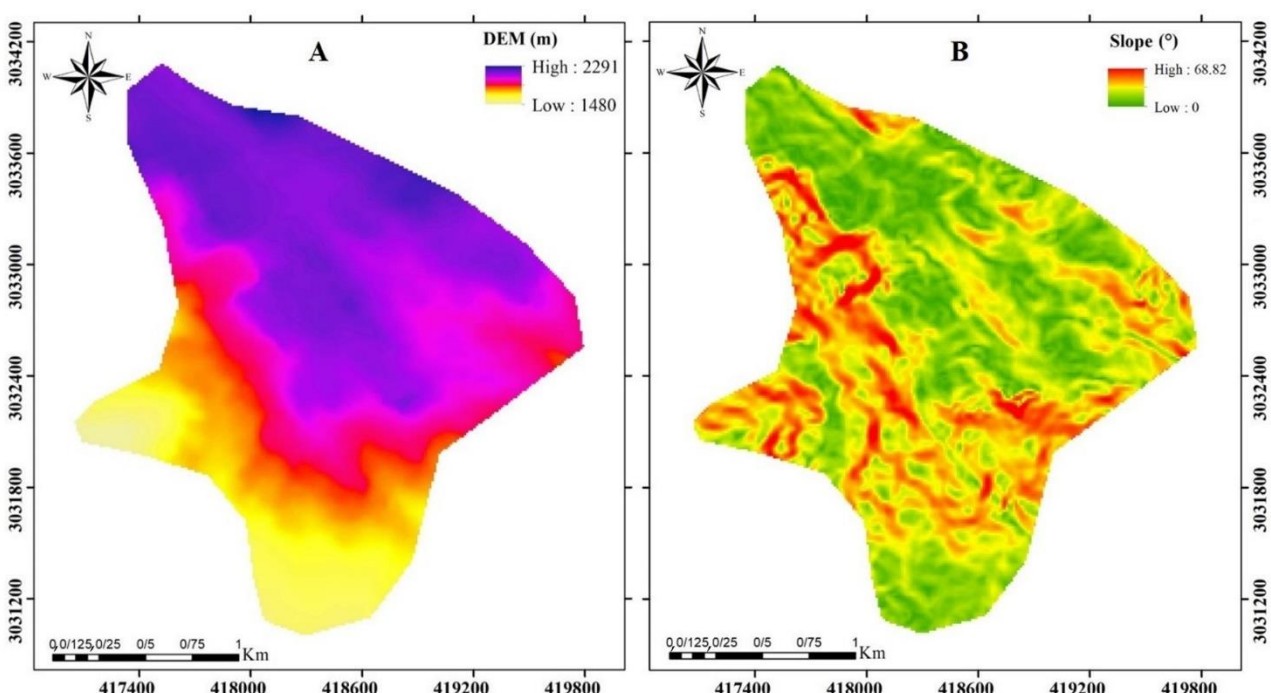

**Figure 1.** Habitat basin and DEM (**A**), and slope degree (**B**).

### 2.2. Assaying the Effects of Topographical and Climatic Factors on Species Distribution

To create species distribution maps, this study was conducted with three stages: (1) generating a species distribution record map of *J. seravschanica* in the natural habitat of Genow, (2) preparation of the dataset, (3) multicollinearity analysis of different independent variables [18]. To generate a distribution roster map of the *J. seravschanica*. At first, the natural habitats of this species were identified in the Genow area. Following that, the location of the presence of 60 trees without dried branches and the location of 70 trees with dried branches of *J. seravschanica* species within 450 ha of the studied site were recorded using extensive field surveys and the Handy GPS app for Android (version 32.6, https://www.binaryearth.net/HandyGPS/index.php (accessed on 10 October 2021)).

Multicollinearity analysis was carried out on four independent climatic factors, including min/max temperatures, wind speed, and annual mean precipitation. The annual mean of four climatic factors was obtained from the Hormozgan Meteorological Bureau (http://www.hormozganmet.ir (accessed on 20 August 2021)). In collinearity analysis, when the variance inflation factor (VIF) is more than or equal to five and tolerance (T) is less than 0.1, then that means that the investigated conditional variables have collinearity [37]. Furthermore, topographical parameters, including slope, aspect, plan and profile curvatures, stream, distance to stream, and elevation, were extracted from ALOS-DEM with 12.5 m × 12.5 m resolution. The DEM was downloaded from the website of the ALOS PALSAR satellite (https://vertex.daac.asf.alaska.edu/ (accessed on 11 January 2022)).

### 2.3. Physiological Evaluation of Population

For investigation into the physiological indices and genetic diversity of the endangered population of *J. seravschanica*, in the natural ecological habitat in Genow, trees were divided into two group: trees with dried branches (A) and trees without dried branches (B) (Figure 2).

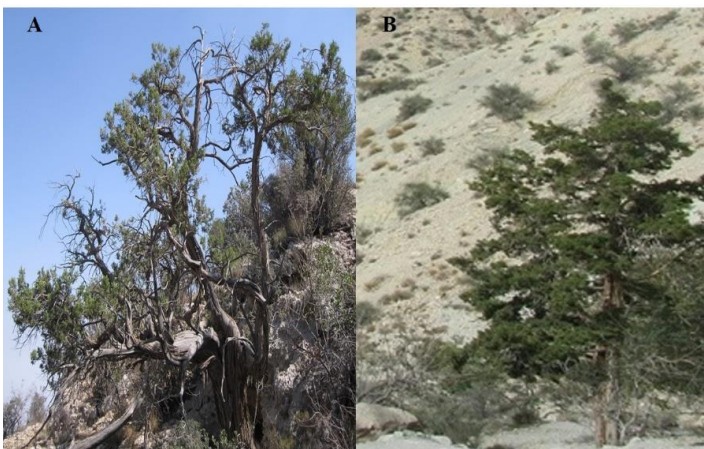

**Figure 2.** Endangered trees in the ecological landscape of Genow; (**A**) tree with dried branches and (**B**) tree without dried branches.

2.3.1. Physiological Assessment

For assessments of the physiological indices, fresh needles were collected from ten trees of *J. seravschanica*, including five trees with dried branches (group A) and five trees without dried branches (group B). Fresh samples were transferred in a polystyrene box to the Laboratory of Physiology, Department of Horticultural Science, Shiraz University, Shiraz, Iran.

The relative water content (RWC) was measured as needle water content according to [38–40]. One gram of fresh needles was weighed (FW) and dried in an oven at 70 °C for 24 h, and then its dry weight (DW) was recorded. The RWC was calculated by the following formula: RWC (%) = (FW−DW)/FW × 100. Relative turgidity (RT) of foliage was measured according to Weatherley [41], and 1 g FW of foliage was weighed and floated on distilled water for 24 h. After floating, samples were removed and quickly surface dried with filter paper and fully turgid (FT) weighed; then samples were dried and their DW measured. Finally, relative turgidity was calculated according to field water content/water content of fully turgid tissue with the following formula:

$$RT\ (\%) = \frac{\text{Field water content (FW} - \text{DW)}}{\text{water content of fully turgid tissue (FT} - \text{DW)}} \times 100 \tag{1}$$

Foliage water deficit (WD) was assessed according to the protocol described by Barrs [42] and Grzesiak et al. [40], and calculated via the formula: WD (%) = 100 − RT. For electrolyte leakage (EL) assessment, an equal amount of shoot tips sample (1 g) was weighed and placed in 40 × 50 mm culture vessels containing 15 mL distilled water and shaken at 80 rpm on an orbital shaker (Parzan Pajouh Shaker, made in Iran) for 24 h at room temperature. The first conductance of the solutions was measured by a conductivity meter, and then culture vessels autoclaved at 115 °C for 10 min and samples were stored at room temperature for 24 h, and, finally, the second conductance of the solutions was measured by a conductivity meter. EL (%) was calculated as initial measurements/final measurements × 100 [43].

Dimethyl-sulphoxide (DMSO) was used to extract chlorophyll and total chlorophyll assessment [44]. For decolorization of leaf tissue, 5 mL of DMSO was added to a tube having 1 g fresh leaf. Tubes were preheated at 65 °C in a water bath for 24 h, and following were cooled at room temperature for 30 min. Next, extract separation and absorbance at 645 nm, and 663 nm measured.

Additionally, proline content was extracted and analyzed according to the method of Bates [45]. Ten mL of sulfosalicylic acid 3% was added to 200 mg of leaf samples and remained for 48 h at room temperature. Then, 2 mL of the supernatant phase of solution was taken and added to ninhydrin and acetic acid (2 mL of each), and heated at 78 °C

in a water bath for 1 h, and following samples were cooled on ice, then toluene (4 mL) was added and vortexed for 20 s. Finally, the supernatant phase containing red complex was used to measure the proline at 520 nm with a spectrophotometer (Shimadzu UV-160A Model, Kyoto, Japan).

### 2.3.2. Assessment of Genetic Diversity

For genetic diversity assessment, samples were collected from two groups of trees (with and without dried branches). Samples were transferred to the laboratory of Tissue Culture and Biotechnology, Department of Horticultural Science, Shiraz University, in liquid nitrogen and stored in a −25 °C freezer until used. DNA was extracted according to the modified cetyltrimethylammonium bromide (CTAB) method [46]. Moreover, twelve ISSR primers were used to recognize the genetic diversity within and among two groups of trees with and without dried branches.

### *2.4. Data Analysis*

The independent-samples *t*-test was used to compare means of physiological indices for two groups of investigated trees, and the correlation of climatic factors and factors collinearity were analyzed by linear regression using SPSS v. 26. For investigation of genetic diversity within and among two groups of *J. seravschanica*, AMOVA analysis was carried out using GeneAleX software, and genetic similarity of evaluating groups assessed with Jaccard's similarity (J) coefficient using the SIMQUAL program in NTSYSpc v. 2.20.

### 3. Results

### *3.1. Topographical Factors' Effects on Species Distribution*

Extracted distribution maps showed that topographical factors have effective roles in the pattern of *J. seravschanica* distribution in the Genow habitat. Elevation especially influenced the distribution of trees with dried branches (Sample_E) and without dried branches (Sample_H) of *J. seravschanica* in the ecological landscape of Genow (Figure 3), and the highest density of species in this habitat was observed at 1950–2100 m ASL. Furthermore, slope and aspect maps indicated that the *J. seravschanica* distribution in the Genow habitat was affected by these two topographic factors. Therefore, this species has greater density on lower slopes, and, in the aspect of northeast, trees were more dispersed (Figure 4). Investigation of streams in the Genow habitat showed that *J. seravschanica* trees were distributed in the areas closer to streams (Figure 5).

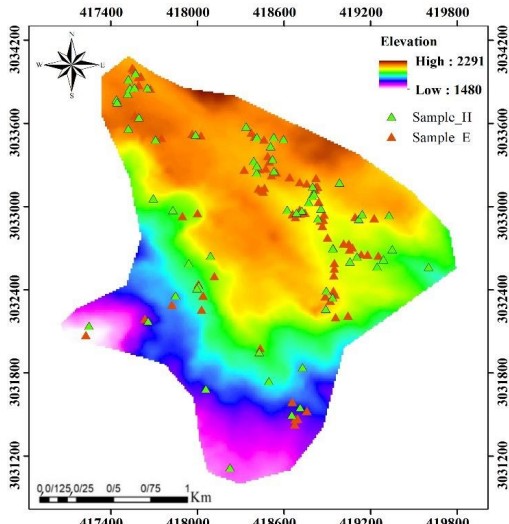

**Figure 3.** Topographical maps of the study area elevation and samples distribution pattern (Sample_H, trees without dried branches; Sample_E, trees with dried branches).

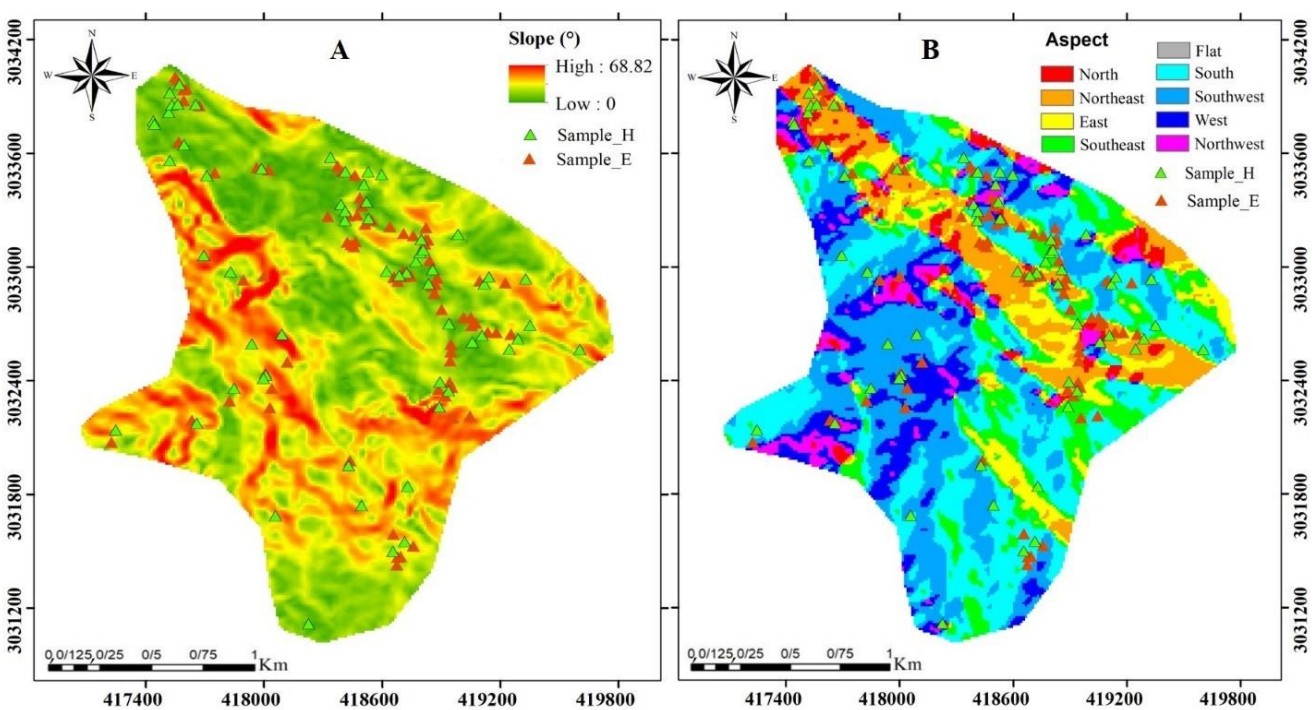

**Figure 4.** Topographical maps of slope (**A**) and aspect (**B**) of the study area and samples distribution pattern (Sample_H, trees without dried branches; Sample_E, trees with dried branches).

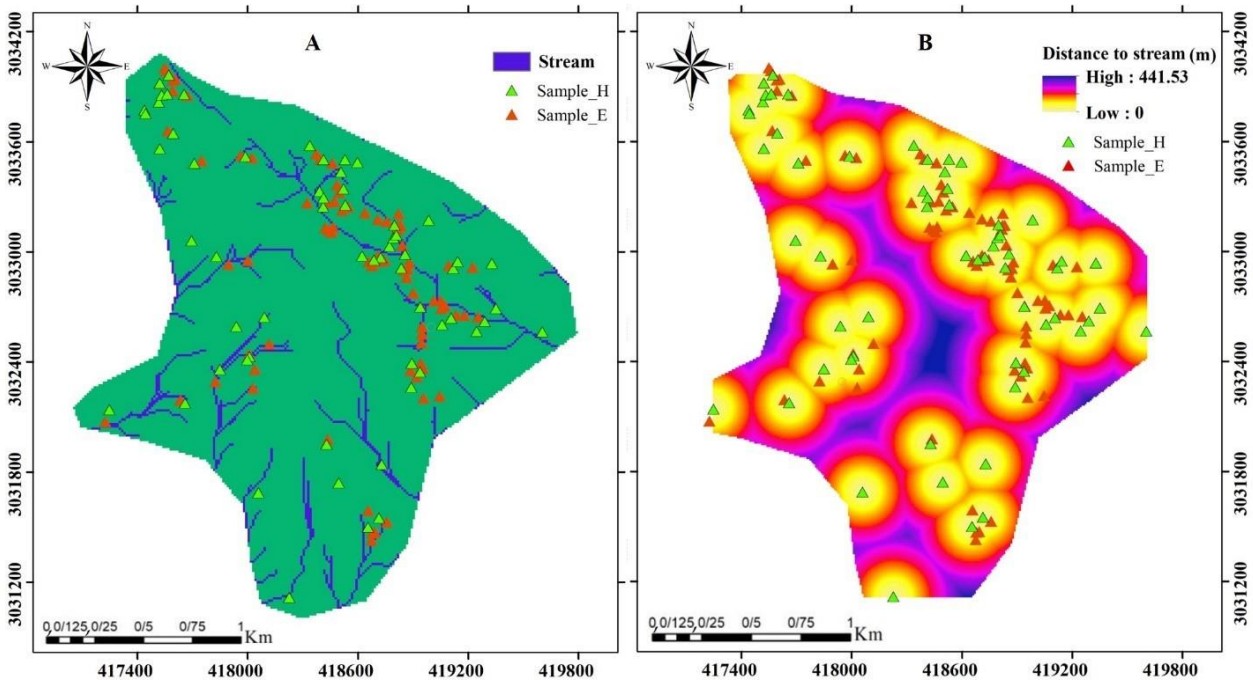

**Figure 5.** Topographical maps of streams in the study area (**A**) and distance (**B**) of investigated samples to stream (Sample_H, trees without dried branches; Sample_E, trees with dried branches).

*3.2. Climatic Factors' Effects on Species Distribution*

Results of linear regression models showed that climatic factors investigated in this study had no collinearity (Table 1). According to these results, the lowest T (0.626) and the highest VIF (2.383) were obtained for precipitation (Table 1).

**Table 1.** Coefficients [†] of multicollinearity of climatic factors with two groups of *J. seravschanica*.

| Model | | Unstandardized Coefficients | | Standardized Coefficients | t | Sig. | Collinearity Statistics | |
|---|---|---|---|---|---|---|---|---|
| | | B | Std. Error | Beta | | | Tolerance | VIF |
| 1 | 1 (Constant) | −14.644 | 4.226 | | −3.466 | .018 | | |
| | Min-Temperature | 0.361 | 0.499 | 0.134 | 0.723 | 0.502 | 0.783 | 1.278 |
| | Max-Temperature | 0.419 | 0.096 | 1.007 | 4.389 | 0.007 | 0.510 | 1.962 |
| | Wind speed | 0.081 | 0.037 | 0.410 | 2.209 | 0.078 | 0.779 | 1.284 |
| | Precipitation | 0.003 | 0.005 | 0.158 | 0.626 | 0.559 | 0.420 | 2.383 |

Note(s): † Dependent Variable: Trees with dried branches (group A), trees without dried branches (group B).

Results of Pearson correlation analysis showed a significant negative correlation between tree groups with max temperature, and there was a significant positive correlation between tree groups with precipitation. Moreover, among the climatic factors, max temperature was negatively correlated with precipitation (Table 2). Correlation analysis of climatic factors and physiological indices showed a positive correlation between max temperature and proline content, WD, and EL of investigated trees. However, the max temperature had a negative correlation with the chlorophyll content, RWC, and RT (Table 2). Also, annual precipitation was negatively correlated with proline content (Table 2). Therefore, an increase in the range of max temperature and reduction of annual precipitation affects the physiological reaction of *J. seravschanica* and influences this species' decline in the ecological landscape of Genow.

**Table 2.** Pearson correlation between climatic factors and physiological indices of two groups of *J. seravschanica* existing in the Genow habitat.

| Variables | Tree Groups | Min-Temp. | Max-Temp. | Wind Speed | Annual Precip. | Chlorophyll | Proline | RWC | RT | WD | EL |
|---|---|---|---|---|---|---|---|---|---|---|---|
| Tree groups | 1 | | | | | | | | | | |
| Min-temp. | 0.108 | 1 | | | | | | | | | |
| Max-temp. | −0.833 ** | −0.341 | 1 | | | | | | | | |
| Wind speed | −0.317 | 0.090 | −0.059 | 1 | | | | | | | |
| Annual Precip. | 0.559 * | 0.413 | −0.652* | −0.283 | 1 | | | | | | |
| Chlorophyll | 0.786 ** | −0.180 | −0.676 * | −0.347 | 0.418 | 1 | | | | | |
| Proline | −0.837 ** | −0.140 | 0.918 ** | 0.182 | −0.640 * | −0.778 ** | 1 | | | | |
| RWC | 0.947 ** | 0.220 | −0.798 ** | −0.290 | 0.413 | 0.668 * | −0.742 * | 1 | | | |
| RT | 0.935 ** | −0.003 | −0.703 * | −0.421 | 0.485 | 0.872 ** | −0.702 * | 0.887 ** | 1 | | |
| WD | −0.935 ** | 0.003 | 0.703 ** | 0.421 | −0.485 | −0.872 ** | 0.702 ** | −0.887 ** | −1.000 ** | 1 | |
| EL | −0.884 ** | −0.081 | 0.709 * | 0.393 | −0.465 | −0.549 | 0.786 ** | −0.865 ** | −0.720 * | 0.720 * | 1 |

Note(s): ** Correlation is significant at 0.01 level (2-tailed). * Correlation is significant at 0.05 level (2-tailed).

Moreover, these results showed a significant correlation between investigated physiological indices and two groups of *J. seravschanica* trees (Table 2). Also, there was a significant positive correlation among chlorophyll content, RWC, and RT in the investigated groups of trees. Furthermore, trees without dried branches (group B) had more chlorophyll content, RWC, and RT, while trees with dried branches (group A) had lower chlorophyll content, RWC, and RT. Therefore, proline content, WD, and EL had a significant negative correlation with groups of trees. Hence, endangered trees with dried branches (group A) had high proline content and more WD and EL than trees without dried branches (group B) (Table 2). Correlation of physiological indices showed that RWC and RT were negatively correlated with WD, EL, and proline content. WD had a positive correlation with EL and proline, but its correlation with chlorophyll content was negative. The EL and proline content had a negative correlation with chlorophyll content, and EL had a positive correlation with the proline content (Table 2).

In addition, an analysis of the relative importance of climatic factors indicated that in the ecological landscape of Genow, max temperature had the strongest effects on *J. seravschanica* declining than other conditional factors, and precipitation was the second strongest factor (Figure 6).

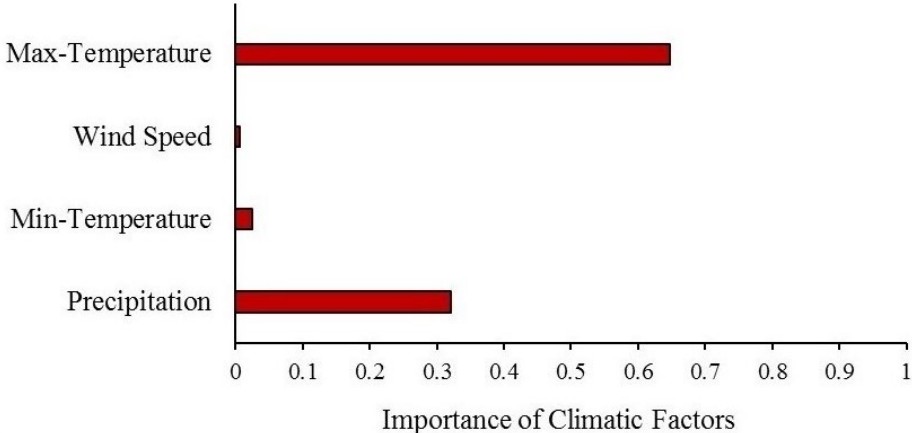

**Figure 6.** Analysis of the relative importance of climatic factors on declining *J. seravschanica*.

### 3.3. Physiological Assessments

Data analysis showed significant differences between two groups of trees for all investigated physiological indices (Table 3). Group statistics showed higher means of proline content, WD, and EL for trees with dried branches than for trees without dried branches (Table 4), while trees with dried branches had a lower amount of chlorophyll content, RWC, and RT than trees without dried branches (Table 4).

**Table 3.** Independent samples *t*-test analysis for physiological indices.

| | Independent Samples Test | | | | | |
|---|---|---|---|---|---|---|
| | Levene's Test for Equality of Variances | | | *t*-Test for Equality of Means | | |
| | | | | | 95% Confidence Interval of the Difference | |
| | t | df | Sig. (2-Tailed) | Mean Difference | Lower | Upper |
| Chlorophyll | −3.597 | 8 | 0.007 ** | 0.26246 | −1.54924 | −0.33876 |
| Proline | 4.330 | 8 | 0.003 ** | 0.27622 | 0.55904 | 1.83296 |
| RWC | −8.303 | 8 | 0.000 ** | 1.26949 | −13.46744 | −7.61256 |
| RT | −7.428 | 8 | 0.000 ** | 0.63008 | −6.13297 | −3.22703 |
| WD | 7.428 | 8 | 0.000 ** | 0.63008 | 3.22703 | 6.13297 |
| EL | 5.347 | 8 | 0.001 ** | 1.02189 | 3.10752 | 7.82048 |

Note(s): ** Two investigated groups of *J. seravschanica* have significant difference at 0.01 level.

**Table 4.** Group statistics for investigated trees of *J. seravschanica*.

| Tree Samples | | N | Mean | Std. Deviation | Std. Error Mean |
|---|---|---|---|---|---|
| Chlorophyll | with dried branches | 5 | 1.9560 | 0.49943 | 0.22335 |
| | without dried branches | 5 | 2.9000 | 0.30822 | 0.13784 |
| Proline | with dried branches | 5 | 2.3900 | 0.60237 | 0.26939 |
| | without dried branches | 5 | 1.1940 | 0.13649 | 0.06104 |
| RWC | with dried branches | 5 | 38.0000 | 2.23607 | 1.00000 |
| | without dried branches | 5 | 48.5400 | 1.74871 | 0.78205 |
| RT | with dried branches | 5 | 93.1800 | 1.21326 | 0.54259 |
| | without dried branches | 5 | 97.8600 | 0.71624 | 0.32031 |
| WD | with dried branches | 5 | 6.8200 | 1.21326 | 0.54259 |
| | without dried branches | 5 | 2.1400 | 0.71624 | 0.32031 |
| EL | with dried branches | 5 | 14.1400 | 1.45162 | 0.64918 |
| | without dried branches | 5 | 8.6760 | 1.76468 | 0.78919 |

*3.4. Populations' Genetic Assessments*

Investigation of genetic diversity for two groups of *J. seravschanica* population existing in the natural habitat of Genow showed significant differences between two groups of investigated trees, with and without dried branches (Table 5). Also, AMOVA analysis showed 83% variation among groups and 17% variation within groups (Table 5). Furthermore, the Jaccard coefficient showed a low similarity between trees with 50% living foliage and trees with 100% living foliage (Figure 7).

**Table 5.** AMOVA analysis of two tree groups' genetic diversity.

| Source | df | SS | MS | Est. Var. | % | PhiPTSstat | |
|---|---|---|---|---|---|---|---|
| | | | | | | Value | *p* (Rand ≥ Data) |
| Among groups | 1 | 84.500 | 84.500 | 16.230 | 83% | 0.829 | 0.007 |
| Within groups | 8 | 26.800 | 3.350 | 3.350 | 17% | | |
| Total | 9 | 111.300 | | 19.580 | 100% | | |

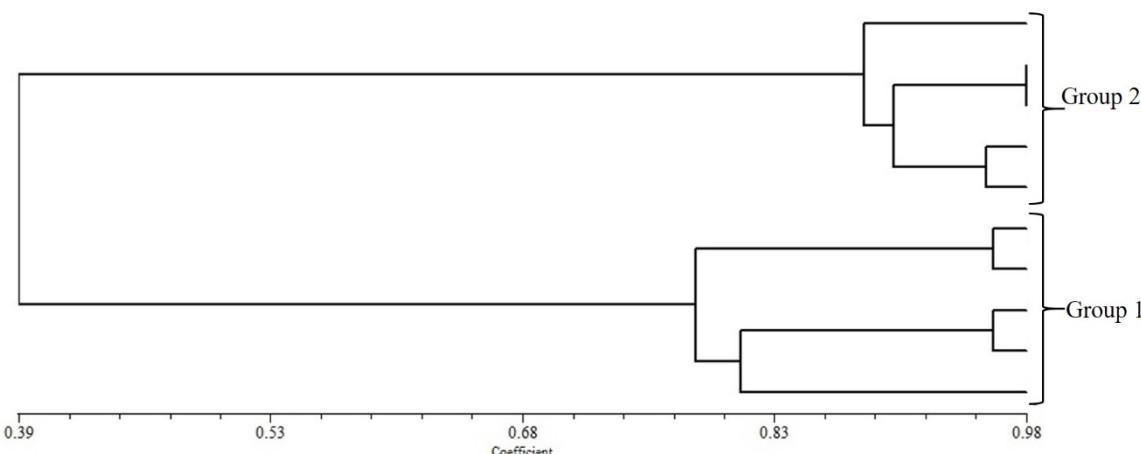

**Figure 7.** Result of the genetic similarity of two groups of *J. seravschanica* (group 1; trees with dried branches), (group 2; trees without dried branches).

## 4. Discussion

This investigation of *J. seravschanica* distribution in the ecological landscape of Genow showed that topographical features had effective roles in *J. seravschanica* distribution. In this regard, the effects of topographic features were confirmed in the restoration of vegetation in a coal mine area [47]. As well, another study investigated the effects of topographic factors on plant distribution and change in several topographic features in the three parallel rivers region of the Tibet plateau, where improvement in the vegetation cover was reported over the period of years from 2000 to 2019 [48]. Therefore, among climatic factors, max temperature had the greatest effects on *J. seravschanica* declining in the Genow habitat. These results conform to the previous study, which showed that *J. seravschanica* has more living foliage in cooler and moister conditions [17]. Therefore, temperature is the most important factor that affects the yield and metabolic activities of the plants' community [49,50]. Moreover, there are interactions between topographical and climatic factors that influence the distribution of plants [51]. Our results showed that *J. seravschanica* had the lowest distribution in high slope and southern aspects. In high slopes and southern aspects, moisture is usually low and temperature is high. Furthermore, according to the negative correlation between temperature and precipitation in the Genow habitat, the increase of max temperature has occurred alongside the decrease in precipitation, and this change significantly influences the viability of *J. seravschanica* in this ecological landscape. This result conforms to the results of previous studies on the effects of heat waves on decrease in frequency and amount of rainfall [52]. Therefore, an increase in max temperature can

influence biodiversity and ecosystems community by two major impacts: first, it influences biodiversity by the pressure of heat stress, and, second, causes precipitation loss and water limitation, which both influences the declining of *J. seravschanica* in the ecological landscape of the Genow habitat. These results are similar to prior studies on climate factors and biodiversity [50,53–55]. In addition, the studies of the species habitat suitability confirmed that max temperature is the most important factor for habitat suitability for the presence of species [18,56,57].

According to the results of the current study, there were differences in investigated physiological indices of two groups of *J. seravschanica* trees: trees with dried branches have a lower RWC of needles than trees without dried branches. This result is consistent with the effects of climate change and drought extension on shedding leaves, limiting the growth and reducing the productivity of woody plants [58–60]. In the Genow habitat, max temperature and precipitation had the greatest effects on the durability of *J. seravschanica*. In this regard, previous studies had shown that extension of drought stress and limitation of soil water significantly affected physiological indices and RWC of Douglas-fir [61,62]. Also, many studies have shown the negative outcomes of drought on forest tree mortality or decline across landscapes [63–65]. Water deficit (WD) is another important parameter that is generally used to investigate the physiological responses of species under climate change and drought stress [66–68]. Hence, in the current study, the trees of *J. seravschanica* with lower living foliage had the highest WD and EL. In this regard, the findings of Poyatos et al. [69] showed that water-deficit in *Pinus sylvestris* L. and *Quercus pubescens* Willd is affected by a combination of two atmospheric and edaphic factors. Therefore, shedding of leaves and growth declining are two main mechanisms occurring in sensitive genotypes of different plants under climate change and drought extension [66,67,70,71]. In the Genow habitat, a decrease of RWC in needles of sensitive genotypes of *J. seravschanica* leads to defoliation and a decrease in total leaves area; these phenomena were stronger for trees with dried branches. As well, the current study showed that leaves of endangered trees with dried branches had high proline content than trees without dried branches. The current result is aligned with the finding of Diamantoglou and Rhizopoulou [72] regarding proline accumulation in the leaf and bark of *Pinus halepensis*. In this regard, many studies have shown water stress causes an increase in the concentration of proline in the leaves of plants [73–75]. An increase in proline is a powerful mechanism of plant cultivar for tolerance to stress conditions [76,77]. Moreover, the results of the current study showed that trees with dry branches have a lower mean of chlorophyll content. Drought is an important phenomena of climate change and abiotic stress that is affecting chlorophyll content [78]. Under drought stress, the leaf water content will decrease and causes reduction in chlorophyll synthesis rate, while the chlorophyll degradation rate is increased [78,79]. So, in the current study, reduction of chlorophyll content in leaves of trees with dry branches occurred due to this genotype sensitivity to drought and climate change. Max temperature and low precipitation in the Genow habitat can reduce photosynthesis and cause starvation of endangered trees that consequently cause the canopy to wither and shoots to dry, finally resulting in tree death. A previous study reported that an increase in abiotic stresses significantly affected forest degradation [80].

Moreover, the genetic assessment showed variation within the populations of *J. seravschanica* that exist in the Genow habitat. Two groups of trees with dried branches and without dried branches, assessed in this study, had significant differences genetically. Therefore, in the populations of *J. seravschanica* existing in the Genow habitat, there are different genotypes, and their vulnerability under climate change depends on genotype characteristics. In this regard, Moran et al. [81] confirmed the relationship between species genetics and their reaction under drought conditions. They reported an effective association between single nucleotide polymorphisms (SNP) and aridity. Other studies reported that the reduction of leaf area by needle abscission and branch death are behaviors of some species of conifers for water loss adjustment under drought and climate change conditions [82,83]. In addition, range shift is the other response of plant species for local adaptation under climate change

and can alter the genetic diversity within species in their natural habitat [84,85]. The current study has shown that genotypes without dried branches have higher stability in the face of harmful climate conditions, and that is promising. Plants have different efficiency in adaptation against stress due to genotypes and genetic diversity [73]. Hence, the *J. seravschanica* trees in the Genow habitat have a logical reaction against climate change according to their genotype, and the genotypes that have no dried branches have high adaptation under max temperature and low precipitation. However, endangered genotypes lose RWC, RT, and chlorophyll content followed by the drying of branches. Consequently, these genotypes may die under intense climate change with high temperatures.

## 5. Conclusions

This investigation of topographical and climatic factors, physiological indices, and population genetic diversity assessment of *J. seravschanica* declining in the ecological landscape of the Genow habitat showed that the distribution of *J. seravschanica* can be influenced by slope degree, aspect, elevation, and distance to streams. Higher temperatures and lower precipitation in the Genow habitat are vulnerable conditions for the Juniperus seravschanica. In response to warming temperatures, the genotypes of *J. seravschanica* with more adaptability have higher durability in the Genow habitat. Moreover, warming continues to be a cause that increases species decline in lower elevations and restricts habitat to higher elevations where conditions are cooler. As well, in response to low precipitation, the *J. seravschanica* habitat is restricted to closer distances to streams. In addition, aspect and slope degree affect the microclimate of trees. The north aspect with low slope is more suitable for growth of *J. seravschanica*. Moreover, in the Genow habitat, max temperature negatively influenced trees' physiological activities and caused the reduction of chlorophyll content, RWC, and RT. Conversely, proline content, WD, and EL increased under max temperature conditions. On the other hand, precipitation is an effective climatic factor that causes the reduction of max temperature and enhances *J. seravschanica's* physiological activities. Consequently, future programs for the cultivation and conservation of *J. seravschanica* should consider the simultaneous effects of slope, aspect, elevation, temperature, and precipitation. Moreover, high genetic variation between two groups of investigated *J. seravschanica* trees and low within-groups diversity logically is related to whether this species declines or endures. Hence, landscape and conservation managers should use asexual propagation of elite genotypes for forthcoming cultivation. These results can be used in the decision-making process for the conservation of *J. seravschanica* in the ecological landscape of Genow or other similar habitats where *Juniperus* are under threat. The results of this study can also be used by resource managers through biodiversity management, in particular when considering the importance of climatic factors for species conservation. Furthermore, the results of genetic diversity are very important for the recognition of elite accessions and decision-making regarding their propagation and future cultivation.

**Author Contributions:** Conceptualization, A.R.B. and E.S.; Methodology, A.R.B. and E.S., Software, A.R.B. and H.S.; Formal analysis, A.R.B., H.S., and E.S.; investigation, A.R.B., H.S. and E.S.; Writing—original draft preparation, A.R.B. and E.S.; Writing—review and editing, A.R.B. and E.S.; Project administration, A.R.B., H.S. and E.S.; Funding acquisition, A.R.B. All authors have read and agreed to the published version of the manuscript.

**Funding:** This study was funded by University of Zabol (IR-UOZ-GR-8952).

**Conflicts of Interest:** The authors declare that they have no conflicts of interest.

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
