# Peer review of "Distribution and Physiology of Juniperus seravschanica Trees in the Genow—The Southernmost and Arid Habitat of Iran"

_water, doi:10.3390/w14213508_

Round 1

Reviewer 1 Report

The paper " Distribution and physiology of Juniperus seravschanica trees in the Genow; the southernmost and arid habitat of Iran " is very good and interesting work that deal with the distribution of endangered trees of J. seravschanica as well as the physiological changes and genetic diversity of these trees in response to climatic factors. For improvements, there are few comments:

1-    The author should correct the affiliation number.

2-    According to journal template, you forgot typing the correspondence as well as e-mails of the rest authors.

3-    Revise and re-write keywords in better words.

4-    Revise all methods. Some of methods should be explained in more details such as DMSO and Proline determination.

5-    You can cite the following recent reference with references 50 and 51 that found in line 260 :- (https://www.mdpi.com/2223-7747/11/9/1219).

6-    Why you did not study the properties of the surrounding soil or did not analyze it ?  

7-    The paper needs English Editing.

Author Response

Dear Professor

Thanks for your comments about improvement of this paper. We eagerly used your comments in revised version of paper and more description around the changes in revised paper is presented at attach.

Thanks for your attention and helpful comments.

Reviewer 2 Report

The article entitled "Distribution and physiology of Juniperus seravschanica trees in the Genow; the southernmost and arid habitat of Iran" aims to evaluate the topographical and climatic influence on physiological indices to understand the distribution of the species Juniperus seravschanica.

The work has merit for publication, however it needs to make the following adjustments.

Abstract:
line 12-13: suggestion: climatic, and genetic factors and physiologic indices on distribution  of J. seravschanica.

line 18: aculiate leaves?

Introduction:

It would be important to address the physiological indices in the introduction. at least one paragraph.

It would also be important to talk about xylem cavitation, as I understand that is the reason for the dry branches.

Materials and Methods

It would have also been important to assess the hydraulic conductivity of the xylem, in order to assess the cavitation level of this tissue, if possible I suggest you do so.

Discussion:

The relationship of the distribution of species with the proximity of streams, reinforces the idea of the relationship with cavitation

I missed discussing the results of proline and its relationship with water stress.

Another point that could be better addressed is the mechanisms that the two genotypes are using in the environment.

What can the lower chlorophyll content in plants with dry branches mean?

Conclusions: 

It has a lot of results.

Look at your goals and build a more objective conclusion.

Author Response

(The authors gave the same response as above.)

Round 2

Reviewer 1 Report

- Accept the modified version after adjusting the following:- Start the method of chlorophyll assessment in a separate paragraph (line 148) as well as proline method (line 152).